

# Validation of the geostrophic approximation using ERA5 and the potential of long-term radio occultation data for supporting wind field monitoring

Irena Nimac[1], Julia Danzer[1], Gottfried Kirchengast[1,2]

[1]Wegener Center for Climate and Global Change, University of Graz, Graz, 8010, Austria
[2]Institute of Physics, University of Graz, Graz, 8010, Austria

*Correspondence to*: Irena Nimac (irena.nimac@uni-graz.at) and Julia Danzer (julia.danzer@uni-graz.at)

**Abstract.** Global long-term stable wind fields are valuable information for climate analyses of atmospheric dynamics. Their monitoring remains a challenging task, given shortcomings of available observations. One promising option for progress is the use of radio occultation (RO) satellite data, based on which the winds are estimated using the geostrophic approximation. Hence, in this study we focus on two goals, explored through European Re-Analysis ERA5 and RO datasets, using monthly-mean January and July data over 2007–2020 with a 2.5° × 2.5° resolution. First, we compare actual and geostrophic ERA5 wind speeds to evaluate the validity of the geostrophic approximation. Second, we test how well ERA5 and RO geostrophic winds agree. We find the geostrophic approximation to work well within 2 m s$^{-1}$ accuracy almost globally (5°–85° latitude), especially over the summer hemisphere; larger differences (more than 5 m s$^{-1}$) may occur in the winter stratosphere. We noticed the effect of large mountain ranges on the wind flow as a wave-like pattern, also in the difference between RO and ERA5 geostrophic winds, pointing to effects of different geopotential height estimations. Generally, RO and ERA5 geostrophic winds showed very good agreement. In the long-term, systematic differences in decadal trends of higher than 0.5 m s$^{-1}$ per decade were found at subtropical latitudes, mainly related to observing system changes in the year 2016 that influenced ERA5. Together with the validity of the geostrophic approximation, this indicates that the long-term stability of RO-derived wind field monitoring can provide added value to reanalysis winds, for the benefit of climate monitoring and analyses.

## 1 Introduction

Accurate wind field information is of great importance in numerical weather prediction, as well as for understanding climate dynamics and chemistry. Such data are regularly assimilated as initial conditions in numerical weather prediction (NWP) models, but also indirectly contribute to advances in climate analysis where such reanalysis data are increasingly used (Stoffelen et al., 2005; Eyre et al., 2020). However, having accurate global 3D wind information still remains a challenging task due to some limitations of specific observation techniques (Stoffelen et al., 2005, 2020). While wind measurements from stations, ships, buoys, or scatterometer winds from satellite radars have generally good spatial coverage over both land and





sea, they lack vertical profile information. On the other hand, various techniques providing vertical profiles such as wind profilers, radio-sounding data, or pilot balloon data have quite coarse spatial coverage. Consequently, over large areas of the southern hemisphere, as well as oceans vertical profiles remain insufficiently known (Stoffelen et al., 2005, 2020).

To overcome these problems between profiling information and good global coverage, altitude-resolving satellite data can be of great help. One such dataset is from the European Space Agency (ESA) Earth Explorer mission Aeolus, which utilizes the

active Doppler Wind Lidar method to measure wind from surface up to 30 km altitude (Stoffelen et al. 2005, Kanitz et al. 2019). It was shown that assimilation of this dataset resulted in improvement of NWP forecasts (Rennie et al., 2021; Žagar et al., 2021). While Aeolus data are useful for better understanding and analysis of atmospheric dynamics such as Kelvin waves (Žagar et al., 2021) or gravity waves (Banyard et al., 2021), due to its quite short time period (launched in August 2018), these data are not suitable for climate change analyses. Another technique to derive vertical profiles is via Global Navigation Satellite

System (GNSS) radio occultation (RO), where the thermodynamic state of the atmosphere is obtained based on the transmitted GNSS radio signals refracted by the Earth's atmosphere (Kursinski et al., 1997; Steiner et al., 2011; Mannucci et al., 2020). The advantage of RO is its unique combination of global coverage, high vertical resolution, high accuracy, long-term stability, and multi-mission data consistency (e.g., Anthes, 2011; Foelsche et al., 2011; Angerer et al., 2017; Zeng et al., 2019; Steiner et al., 2020a). The RO data sets are assimilated into operational weather forecasts (e.g., Healy and Thépaut, 2006; Buontempo

et al., 2008; Cardinali 2009) and long-term reanalyses (e.g., Hersbach et al., 2020; Kobayashi et al., 2015; Gelaro et al., 2017), and are used for climate analysis studies (e.g., Steiner et al., 2011, 2020b; Stocker et al., 2021).

While RO does not directly provide wind information, winds can be derived from geopotential information using the geostrophic approximation. Even though the geostrophic approximation is widely used to derive dynamics from satellite information based on the mass (geopotential height) field (e.g., Elson 1986; Oberheide et al., 2002; Scherllin-Pirscher et al.,

2014, 2017; Verkhoglyadova et al., 2014), its accuracy and validity for different latitudinal and altitudinal regions, as well as region of breaking-down, has not been thoroughly investigated. To test the validity of geostrophic approximation, one needs to use a dataset which contains information on both the pressure-geopotential height relation (thermodynamics), needed to estimate geostrophic winds, and the actual wind (dynamics). Such validation studies were already made decades ago using measurements such as rawinsonde (e.g., Wu and Jehn 1972) or climate models (e.g., Boville 1987; Randel 1987).

Boville (1987) tested the validity of the geostrophic approximation using a general circulation model from lower troposphere to the stratopause region of the Northern hemisphere in January and showed that errors are much larger in the stratosphere region compared to the troposphere. Randel (1987) comments that overestimation of geostrophic wind speed in the high-latitude stratosphere of the winter hemisphere is primarily a result of neglecting curvature effects, while Elson (1986) mentions that increased wave activity related to sudden warming mainly contributes to producing ageostrophic effects. As a dataset

which joins advantages of both climate models in terms of spatial and temporal coverage and measurements in terms of accuracy, state-of-the-art reanalysis data neatly provide the necessary information to test the validity of the geostrophic approximation.



Besides a seasonal and altitudinal dependence on the validity of the geostrophic approximation, another limitation is its breakdown towards the equatorial region as the Coriolis parameter approaches zero. Oberheide et al. (2002) linearly

interpolated geostrophic wind fields between ±10°, while Scherllin-Pirscher et al. (2014) and Verkhoglyadova et al. (2014) left out the regions of ±15° and ±10°, respectively, from their wind estimation studies. To our knowledge, there are no up-to-date studies dealing with a rigorous evaluation of the geostrophic approximation with focus on climatological long-term wind field monitoring. This is especially important in regard to recent improvements in both measurements and climate models in terms of finer temporal and spatial resolution, as well as parametrizations and processes included (Rummukainen, 2010).

Hence, in this study we focus on two main goals. First, we use reanalysis data to test the validity of the geostrophic approximation for representing monthly-mean winds, as a method potentially vitally helpful for deriving long-term changes in dynamics (wind fields) from long-term thermodynamic satellite data (mass fields). Second, we evaluate the utility of RO-derived monthly-mean winds, for their potential added value as a separate wind field monitoring data record providing improved long-term stability. The paper is structured as follows: In Sect. 2 we describe data and method used in the study. The

results are presented in the Sect. 3, while Sect. 4 covers the discussion part. Conclusions and perspectives are finally given in Sect. 5.

## 2 Data and study method

In this analysis we used global monthly-mean ERA5 reanalysis (Hersbach et al., 2020) and multi-satellite RO OPSv5.6 data (Angerer at al., 2017; Steiner et al., 2020a) during the joint time period 2007 to 2020. We analysed the global wind data at 2.5°

latitude × 2.5° longitude resolution, in the altitude region from near surface (1000 hPa) up to the middle stratosphere (10 hPa, about 32 km). We chose January and July as two representative months for the winter and summer season. A further advantage of those two months is that the strongest wind speeds in jet stream regions are observed (e.g., Scherllin-Pirscher et al., 2014, 2017), reaffirming that these serve as adequate test data for the focus goals of this study. For both months, we calculated the long-term monthly-mean wind speed fields over the 14-year period of 2007 to 2020.


### 2.1 ERA5 reanalysis data

As a state-of-the art reference dataset to test the validity of the geostrophic approximation, we used the European Re-Analysis 5th Version (ERA5) of the European Centre for Medium-Range Weather Forecasts (ECMWF). On the chosen latitude-longitude grid of the selected monthly-mean data, we extracted eastward-wind and northward-wind components, for computing

the actual wind speeds, as well as isobaric geopotential height data (geopotential fields on pressure levels), for deriving the geostrophic winds. We term the actual wind speeds $ERA_o$ (for "ERA5 original winds") and the geostrophic ones $ERA_g$ (ERA5 geostrophic winds), respectively.



**2.2 RO satellite data**

The RO multi-satellite climatologies are derived from the satellite missions CHAMP (Wickert et al., 2001), C/NOFS (de la Beaujardiere et al., 2004), F3C (Anthes et al., 2008), GRACE (Beyerle et al., 2005; Wickert et al., 2005), MetOp (Luntama et al., 2008), and SAC-C (Hajj et al., 2004). Phase data were derived at UCAR/CDAAC (University Corporation for Atmospheric Research/COSMIC Data Analysis and Archive Center), and further processed at the Wegener Center (WEGC) using the Occultation Processing System OPSv5.6 (Angerer et al., 2017; Steiner et al., 2020a). Monthly $2.5° \times 2.5°$ gridded data were

then derived using aggregated atmospheric profile data weighted according to the longitude-latitude distances of each profile to the bin centre using Gaussian-radius-lon-lat weighting within a radius of 600 km. On average, the number of RO profiles is around 60 000 profiles per month. In order to derive the RO geostrophic wind speeds, we used isobaric geopotential height data (as for ERA5) and term these RO-derived wind speeds $RO_g$ (RO geostrophic winds).

RO data show high quality and high vertical resolution of the relevant isobaric geopotential height fields over the altitude

region of 5 km to 35 km (Scherllin-Pirscher et al., 2017; Steiner et al., 2020a), though in the moist troposphere background information of (re)analysis data supports the thermodynamic data retrieval from atmospheric refractivity (Scherllin-Pirscher et al., 2017; Li et al., 2019). Towards higher altitudes into the upper stratosphere, the impact of residual errors due to measurement noise and ionosphere starts to increase (e.g., Danzer et al., 2013, 2018; Liu et al., 2018), decreasing the accuracy of the retrieved isobaric geopotential height data. Hence, we focused our evaluation of RO utility up to the middle stratosphere.

**2.3 Study method**

We studied the validity of the geostrophic approximation (first goal) as the difference of $ERA_g$ and $ERA_o$ wind fields and evaluated the deviations of the observationally derived geostrophic winds from the reanalysis-derived ones (second study goal) in terms of the $RO_g$ vs. $ERA_g$ difference (Fig. 1). We inspected vertical cross-sections of the respective wind (speed) field differences from the near-surface troposphere levels up to mid-stratosphere. Complementary, for inspecting horizontal latitude-

longitude maps, we concentrated on the three representative levels 200 hPa, 50 hPa and 10 hPa, which represent the tropopause, lower stratosphere and middle stratosphere regions, respectively. Additionally, we chose four smaller geographical regions for a more detailed look on the validity of the geostrophic approximation. The areas represent two regions in the Southern (SH) and Northern Hemisphere (NH) where the geostrophic approximation was found accurately valid (areas in the South and North Pacific Ocean) and two, where we found comparatively large differences (Himalayan and Andes region).



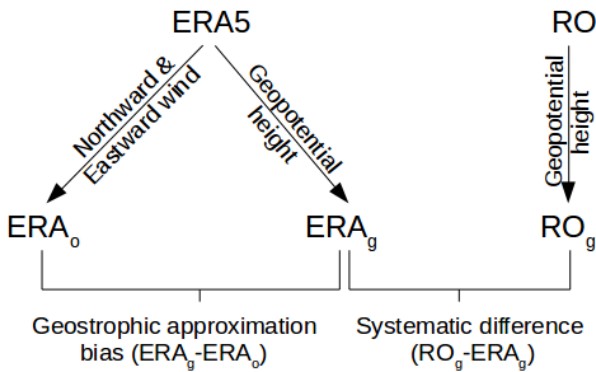

**Figure 1:** Schematic diagram of the two-steps evaluation method. ERA$_o$ stands for the original (actual) ERA5 wind speeds, while ERA$_g$ and RO$_g$ stand for geostrophic winds, respectively.

To derive wind fields, one commonly starts with the equations of zonal and meridional momentum (Holton, 2012). However, due to the complexity of solving these nonlinear partial differential equations, some assumptions and simplifications are useful to estimate approximate wind components. To derive them in line with the focus of this study from thermodynamic data such as provided by RO, we use the geostrophic approximation. In this approximation, most of the horizontal momentum equation terms are neglected, except for the Coriolis force term, which is retained and balances the pressure gradient force. In the isobaric coordinate system, zonal ($u_g$) and meridional ($v_g$) geostrophic wind components are given by the relations (Holton, 2012; Scherllin-Pirscher et al., 2014; Verkhoglyadova et al., 2014):

$$u_g = \frac{-1}{f(\varphi)a} \frac{\partial \Phi}{\partial \varphi}, \tag{1}$$

$$v_g = \frac{1}{f(\varphi)acos(\varphi)} \frac{\partial \Phi}{\partial \lambda}, \tag{2}$$

where f($\varphi$) is the Coriolis parameter, f($\varphi$) = 2 $\Omega$ sin$\varphi$, with $\Omega$ = 7.2921×10$^{-5}$, a is the Earth's radius, $\Phi$ denotes geopotential on isobaric levels, $\varphi$ is geographic latitude, and $\lambda$ longitude. Geopotential $\Phi$ is calculated as $\Phi = Z\,g_0$ where $Z$ is geopotential height and $g_0$ = 9.80665 m s$^{-2}$ the standard gravity constant. Hence, to derive geostrophic wind fields, we need geopotential height fields at pressure levels as information.

ERA5 geostrophic wind was calculated based on the corresponding geopotential field resulting in wind fields over the region ±87.5° latitude. Since deriving RO geostrophic wind fields is based on the horizontal derivatives of geopotential height, we first smoothed the 2.5° × 2.5° geopotential fields using a 5-point Gaussian filter in both longitudinal and latitudinal direction, to avoid high spatial variability of the wind fields. In latitudinal direction, the last two latitude circle grid lines were excluded from the analysis, since they are needed as filter margin. Additionally, one more grid line (85°) was discarded after calculating





the derivative according to Eq. (1). The final latitudinal range used for the RO-derived fields is ±82.5°. Related to this, due to the lower number of soundings over the polar caps, it is justified to exclude these few polar latitude circles from the analysis. As Coriolis parameter $f(\varphi)$ approaches zero near the equator, the approximation is not valid in those areas. Still, as in our data

grids the lowest latitude bin grid lines are at ±1.25° latitude, it was possible to calculate geostrophic wind for all climatological bins, though values close to the equator lose physical meaning. This way we determined the region of geostrophic approximation break-down by comparing geostrophic approximation bias to some commonly used accuracy requirement values.

We used the monthly-mean geopotential data at isobaric levels for the January and July months in the period 2007–2020 to

derive the geostrophic wind components using equations (1) and (2), and subsequently computed the speed as the magnitude of the corresponding wind vector. The wind speeds for $ERA_g$, $ERA_o$, and $RO_g$ were then used to perform our evaluations according to Fig. 1. All calculations, statistical analysis and visualization were performed using Python programming language and mainly its packages numpy, xarray, pymannkendall and matplotlib.

To put the results into context with wind accuracy requirements (e.g., Stoffelen et al., 2020), we used absolute requirement

values for domains with small wind speeds, while for large wind speeds relative requirement values appeared a more reasonable choice. As target requirement, we chose a difference of ±2 m s$^{-1}$ or a relative difference of ±5 %, while maximum threshold values were set to ±5 m s$^{-1}$ or ±10 %, respectively. These requirement values are chosen to be consistent with the wind observation accuracy requirements specified by the World Meteorological Organization (WMO) for various applications, including NWP and climate (Stoffelen et al., 2020; WMO-OSCAR, 2022; Table 1). Exceeding the threshold requirements is

taken, in our evaluation, to signal a breaking-down of the geostrophic approximation. Simple linear regression was used to test the long-term temporal stability of the derived wind speed fields. We estimated the decadal trend rate in $RO_g$, $ERA_g$, and their difference, and evaluated this against the WMO-GCOS (2016) wind measurement stability target requirement of ±0.5 m s$^{-1}$ per decade (see also Table 1).

**Table 1:** Selected absolute and relative wind speed accuracy requirements used in the study, informed by WMO-GCOS (2016).

| Accuracy Specifications | Wind speed metric | Target requirement ("high accuracy") | Threshold requirement ("marginally good") |
|---|---|---|---|
| Validity of geostrophic approximation | Absolute (m s$^{-1}$) | ±2 | ±5 |
| | Relative (%) | ±5 | ±10 |
| Temporal stability check | Absolute (m s$^{-1}$ per decade) | ±0.5 | - |





## 3 Results

### 3.1 Validity of geostrophic approximation – ERA$_g$ vs. ERA$_o$

To test the validity of the geostrophic approximation, ERA$_g$ is compared to ERA$_o$ winds. In January, the geostrophic
approximation is valid within ±2 m s$^{-1}$ poleward from ±5° latitude in both hemispheres, mostly through all observed altitude
levels (Fig. 2a). The exceptions are the mid- stratospheric altitudes at middle latitudes in the winter hemisphere, where the bias
is still below about 6 m s$^{-1}$, however. Larger differences are also found at the lowest (boundary layer) levels in the NH mid-
latitudes, as well as in the Antarctic. In July, the validity of approximation within 2 m s$^{-1}$ is also generally observed down to
±5° latitude for both hemispheres, with the exception of larger bias at the upper altitude levels at winter hemisphere mid-
latitudes and near-surface regions over the Antarctic ice shield, where this difference reaches more than 10 m s$^{-1}$ (Fig. 2b).
The larger wind speed differences at the stratospheric levels of the winter hemisphere correspond to the polar jet stream region.
The amplitude of differences is larger in July when the jet stream is also stronger. It can be noticed that in both seasons, the
subtropical jet stream is well represented by the geostrophic approximation.

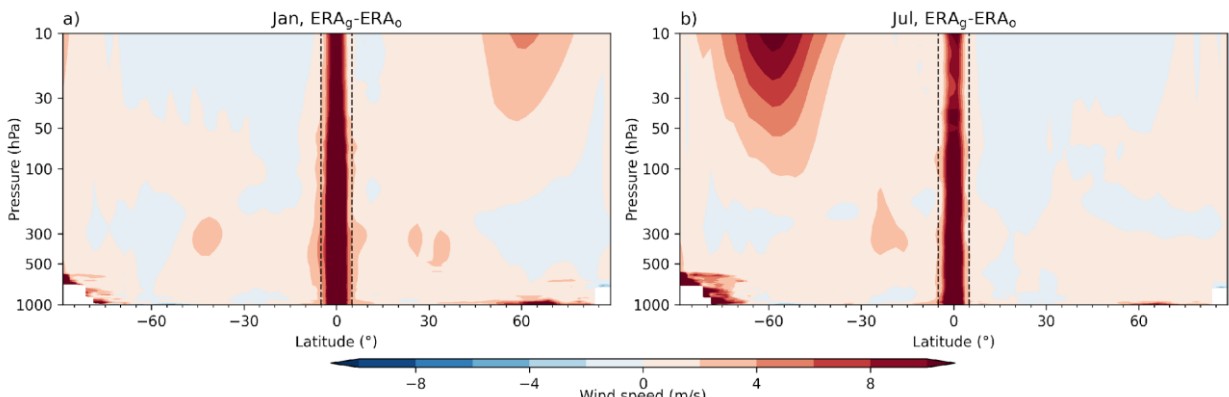


**Figure 2:** Long-term (2007–2020) mean vertical cross-section of wind speed difference between ERA5 geostrophic wind
(ERA$_g$) and ERA5 original wind (ERA$_o$) for January (left) and July (right). Dashed black vertical lines delineate the ±5°
latitude band around the equator.

Even though larger differences in the troposphere are mainly located around the equator according to the vertical cross sections
(Fig. 2), in the horizontal wind field maps at representative pressure levels we also found interesting features above large
mountains and obstacles such as the Himalaya, Andes, and Greenland (Fig. 3). It can be seen that above Greenland and the
Himalaya, the amplitude of these patterns is larger in January when wind speeds are stronger than in July (Fig. 3a, b). The
patterns are also noticeable as high up as the mid-stratospheric 10 hPa level (Fig. 3e, f). Larger biases in higher latitude areas
at the 10 hPa level correspond to the location of the polar jet streams.



**Figure 3:** Differences in the long-term (2007–2020) mean geostrophic ERA5 (ERA$_g$) and original ERA5 (ERA$_o$) wind speeds at the 200 hPa (top), 50 hPa (middle) and 10 hPa (bottom) level, for January (left) and July (right). Areas with differences larger than 10 m s$^{-1}$ are delineated with black contour lines. Green rectangles highlight four selected regions for which more detailed inspections were made. Dotted black horizontal lines delineate the ±5° latitude band around the equator.

To validate the geostrophic approximation bias in regard to specific WMO-related requirements (Table 1), we inspected the results in more quantitative detail (Fig. 4). At all three representative pressure levels in both seasons (months), the largest differences exceeding the requirements are observed in the ±5° equatorial band. At 200 hPa both in January (Fig. 4a) and July (Fig. 4b), as well as at 50 hPa in January (Fig. 4c), differences within all the remainder of the domain are within the target





requirement of 2 m s⁻¹. At 50 hPa in July (Fig. 4d) and at 10 hPa in January (Fig. 4e) somewhat larger biases are found at middle latitudes of the winter hemisphere, but still within the threshold requirement of 5 m s⁻¹. Only at the highest 10 hPa level in July (Fig. 4f), a larger bias at SH mid-latitudes exceeds also the threshold requirement.

**Figure 4:** Geostrophic approximation bias vs. latitude (black line) at the 200 hPa (top), 50 hPa (middle) and 10 hPa (bottom) level, for January (left) and July (right). WMO-based accuracy requirement values are indicated at values of 2 m s⁻¹ (ABSrv2, light blue line) and 5 m s⁻¹ (ABSrv5, dark blue) and, in relative terms, at 5 % (RELrv5, orange) and 10 % (RELrv10, red).

We next inspected the NH and SH domains without the ±5° equator band and the four focus regions highlighted in Fig. 2 in quantitative detail (Fig. 5). Two of the regions are above the Himalaya and Andes mountains, where we see an oscillatory pattern in the difference. At the lowest level of 200 hPa (Fig. 5a), in both seasons and both hemispheres as well as in the focus



regions over the northern and southern Pacific, the median of geostrophic bias is all over close to zero, while in the Himalaya and Andes regions, the geostrophic approximation shows a tendency to overestimate the wind speeds, in particular in winter. At 50 hPa (Fig. 5b), some overestimation of the speeds by the geostrophic method is seen in winter, while in summer the

215 estimates appear unbiased. Geostrophic bias larger than the target requirement of 2 m s$^{-1}$ is seen over the Himalaya and Andes in winter, while in summer it is close to zero. At the 10 hPa mid-stratosphere level (Fig. 5c), the bias over both hemispheres in winter is even larger, especially over the SH where it exceeds the 2 m s$^{-1}$ target requirement. In both seasons, the median bias over both Pacific regions is found within the ±2 m s$^{-1}$ target range, while over the mountain regions it is exceeding this range.

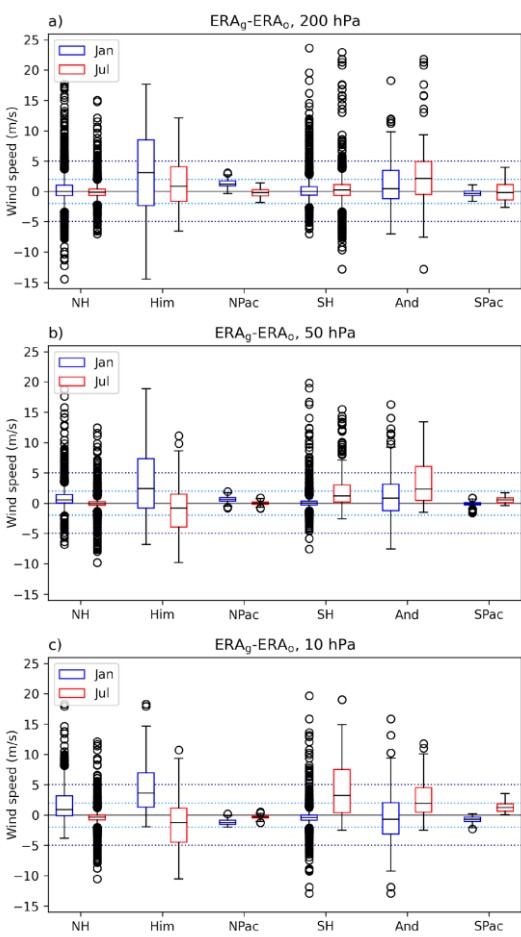

**Figure 5:** Overview statistics of geostrophic approximation bias over the Northern and Southern Hemisphere (NH, SH) and over four selected regions: Himalayan (Him), Andes (And) and parts of the North Pacific (NPac) and South Pacific (SPac), for January (blue) and July (red) at a) 200 hPa, b) 50 hPa and c) 10 hPa levels. Horizontal dashed lines indicate the target and threshold requirements, respectively, summarized in Table 1.



Besides the median estimate of the bias, differences are also noticeable in its variability around the median in selected regions. In particular, the biases above the two Pacific regions exhibit much lower spread compared to the ones over the two mountain ranges.

## 3.2 Utility of RO-derived winds – RO$_g$ vs. ERA$_g$

In line with the second study goal, we tested how well RO and ERA5 geostrophic wind field data agree. Here, the equatorial band within ±5° and the polar caps (poleward of 82.5°) are already excluded from further inspections, given the results in Sect. 3.1. That is, we focus on understanding the consistency between RO-derived and ERA5 geostrophic winds in those (still) nearly global domains, where the geostrophic approximation is found to perform well.

Vertical cross-sections of the difference between two datasets generally show good agreement in both seasons and over troposphere and stratosphere, with average differences of around ± 2m s$^{-1}$ (Fig. 6). The main exception are higher values at the lower troposphere heights, especially over the SH polar region in July.

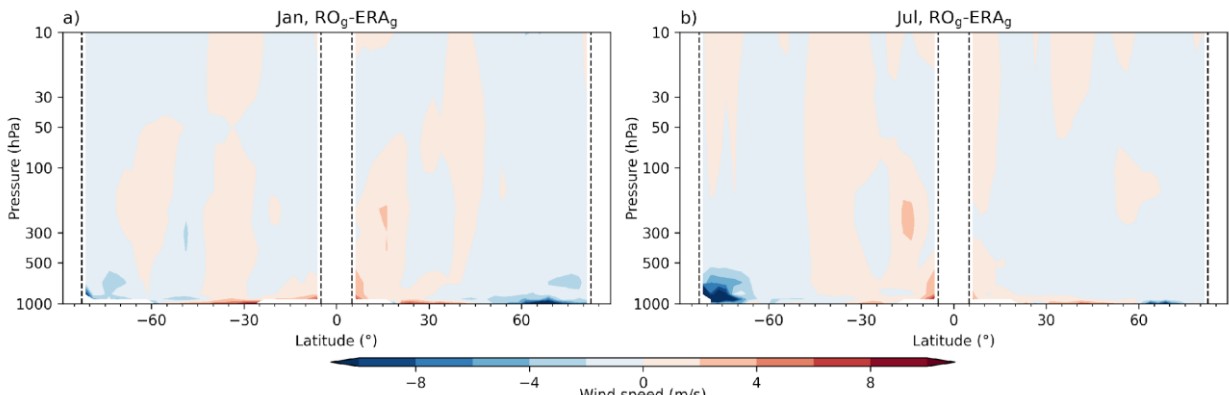

**Figure 6:** Long-term (2007–2020) mean vertical cross-sections of wind speed differences between RO geostrophic wind (RO$_g$) and ERA5 geostrophic wind (ERA$_g$), for a) January and b) July. Black dashed vertical lines delineate the ±5° equator band and the polar caps beyond ±82.5°.

Inspection of the differences in latitude-longitude maps at selected levels reveals that the overall patterns of the subtropical and polar jet streams are well represented also by the RO data, but oscillatory patterns above the large mountain ranges are also for this dataset (Fig. 7). Generally, the differences are larger in the winter hemisphere, especially at 200 hPa (Fig. 7a, b). At 50 hPa, differences larger than ±2 m s$^{-1}$ are seen in the wave-pattern structures in both seasons above the Himalaya and Andes regions (Fig. 7c, d). In January, at the highest level of 10 hPa, besides larger differences above the mentioned mountains, differences up to –4 m s$^{-1}$ are found at high latitudes, corresponding to the position of polar jet stream (Fig. 7e). In July, such differences in the polar jet stream region are not seen (Fig. 7f).



**Figure 7:** Difference in the long-term mean (2007–2020) geostrophic RO (RO$_g$) and geostrophic ERA5 (ERA$_g$) wind speeds at 200 hPa (top), 50 hPa (middle) and 10 hPa (bottom), for January (left) and July (right). The western Pacific longitude sector indicated by green lines in a) and b) is used for the subsequent analysis of the long-term temporal consistency of datasets. Black dashed horizontal lines delineate the ±5° equator band and the polar caps beyond ±82.5°.

Additionally, we investigated the long-term temporal consistency of the two datasets and show results in terms of differences found in the 140° E - 160° E western Pacific longitude sector, at the 200 hPa tropopause-region level. This longitude sector was selected, because the subtropical jet stream seems to leave a quite distinct feature over the western Pacific; differences of





up to ±6 m s⁻¹ (i.e., up to exceeding the Table 1 threshold requirements) are seen in this sector in the winter hemisphere (see

Fig. 7a, b).

A 2007–2020 temporal analysis (Fig. 8) reveals that this pattern is systematic, with belts of positive and negative differences within 10° to 40° latitude in the winter hemisphere (Fig. 8, left). We find a noticeable decrease in the amplitude of the systematic difference after the year 2016. This coincides with a major observing system change in the ERA5 data assimilation (Hersbach et al., 2020; Fig. 4 therein). To better discern effects of temporal change, we specifically inspected the 10°–20°,

20°–30° and 30°–40° latitude band results, including the inspection of possible trends in the changes (Fig. 8, right). The RO-ERA5 difference evidently appears to decrease in some of the bands, while in other bands it is quite long-term stable (Table 2). Exceedance of the WMO-GCOS target requirement for long-term stability within ±0.5 m s⁻¹ per decade, taken as a consistency benchmark, occurs in January for the 20°–30° N and 30°–40° N bands and in July for the 10°–20° S and 20°–30° S bands, respectively.

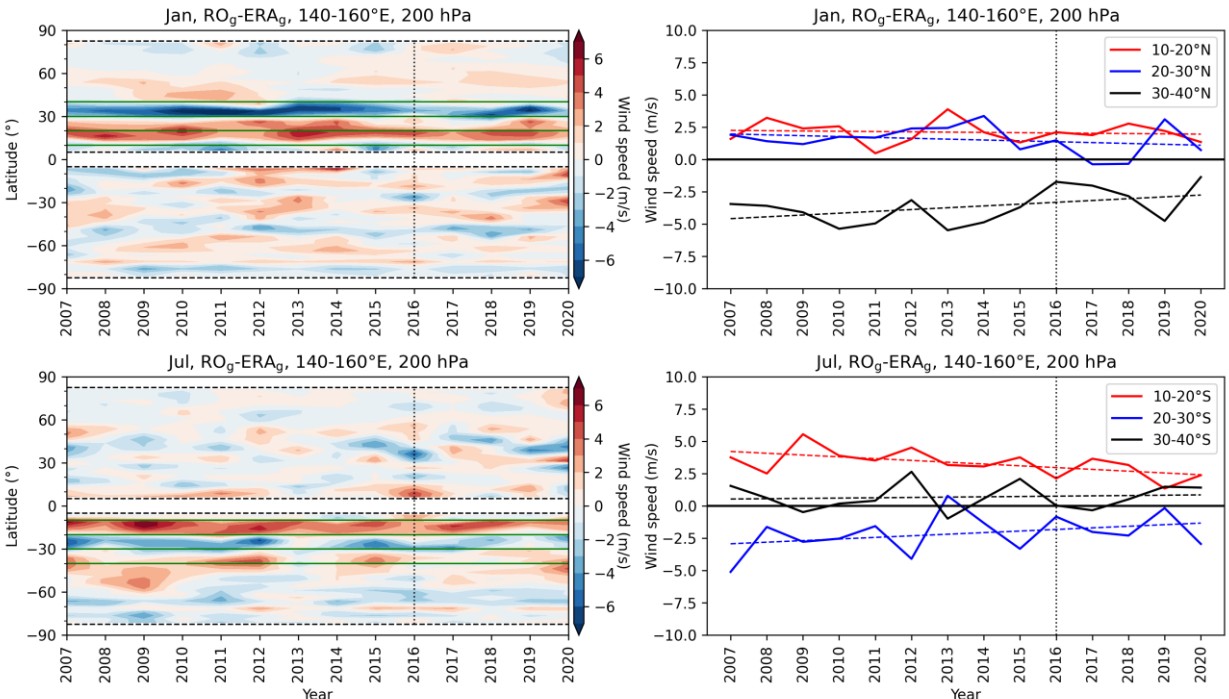

**Figure 8:** Latitude-resolved time series of the difference in RO geostrophic (RO$_g$) and ERA5 geostrophic (ERA$_g$) wind speeds at 200 hPa, for January (top) and July (bottom) over the area 140° E–160° E longitude sector (left column). The right column shows the corresponding annual 10°-band averaged RO-ERA5 difference time series for the winter-hemisphere 10°–20°, 20°–

30° and 30°–40° latitude bands (marked by green horizontal lines in the left panels), plus the associated linear-fit trend lines for which the trend rates are summarized in Table 2. Black dashed horizontal lines delineate the ±5° equator band and the polar caps beyond ±82.5° and black dotted vertical lines mark the year 2016, where ERA5 saw a change in observing systems.





**Table 2:** Decadal trend rates (m s$^{-1}$ per decade) in the difference between RO and ERA5 geostrophic wind speeds over 2007
to 2021 in selected winter-hemisphere latitude bands in January and July for the western Pacific longitude sector 140° E–160°
E. RO-ERA5 difference trend rates larger than the WMO-GCOS (2016) long-term stability requirement of ±0.5 m s$^{-1}$ per
decade (Table 1) are bold-faced.

| Latitude band | January trends NH (m s$^{-1}$ per decade) | July trends SH (m s$^{-1}$ per decade) |
|:---:|:---:|:---:|
| 10°–20° | -0.21 | **-1.29** |
| 20°–30° | **-0.62** | **1.14** |
| 30°–40° | **1.30** | 0.23 |

## 4 Discussion

When evaluating the validity of the geostrophic approximation, we found a clear seasonal distinction with larger differences
in the winter hemisphere compared to the summer. This was quite expected, due to more unstable weather conditions in winter
compared to the summer season, when the ageostrophic contributions are less expressed (Wu and Jehn 1972; Scherllin-Prischer
et al., 2014, 2017; Verkhoglyadova et al., 2014). This is one of the reasons why early geostrophic validation studies a few
decades ago were mainly performed for the winter season (e.g., Elson 1986; Boville 1987; Randel 1987).

Through analysing altitudinal differences, we found better agreement in the troposphere, compared to the stratosphere region,
probably due to a larger contribution of curvature effects and related gradient winds (Randel, 1987; Boville, 1987). Unlike in
other studies, here we detected the effect of larger obstacles on the wind flow as a part of the geostrophic bias. These wave-
like patterns are likely associated with orographic gravity waves (e.g., Smith, 1982). The characteristics of the gravity waves
over large mountain ranges depend on the mountain dimensions (height, width and length) and its orientation in regard to wind
flow, static stability profiles as well as on the mean wind speed (e.g., Holton, 2012). Evidently, the combination of the large
topographic obstacles and the dynamic jet-stream above it, results in quasi-stationary pressure/geopotential patterns above
these mountains which are then also still fingerprinted into monthly-mean fields. Hence, the geostrophic approximation is
somewhat less accurately valid above larger mountains. Depending on the resolution of the dataset used, this wave-like pattern
might not be seen in coarser spatial grids, due to averaging out the pattern over (even) larger areas to close to zero.

The analysis of the utility and potential added value of RO-derived winds, from inspecting the difference between RO and
ERA5 geostrophic winds, revealed generally good agreement over the whole near-global troposphere-stratosphere domain
where the geostrophic approximation was found robustly valid. Still, wave-like patterns above large mountain ranges were
also present in the difference of these two fields. We find evidence that this mountain effect comes from ERA5 data, since
ERA5 geostrophic wind field maps at pressure levels exhibit such wavy behaviour while RO-derived ones do not. Even though
the geopotential difference between the two data sets is quite small, compared to the magnitude of the geopotential itself (below
1 %), we find that such small differences can lead to appreciable differences in wind speeds (more than 10 m s$^{-1}$), since these





derive from the spatial derivatives of the isobaric geopotential fields. Hence the question arises: why do we see this (indirect) gravity wave effect in geopotential fields, and consequently in geostrophic winds, in ERA5 but not in RO?

One of the possible sources of these differences in geopotential may be due to the way these are constructed. In ERA5, vertical pressure integration is performed bottom-up, resulting in vertical propagation of larger errors from the (lower) troposphere region. In contrast, RO-derived data work with downward vertical integration from the tenuous mesosphere, so that pressure integration is more robust and leading to smaller errors (Leroy, 1997; Scherllin-Pirscher et al., 2017). Hence, besides high vertical resolution, this is an advantage of RO, related to its "power of vertical geolocation" as described in detail by Scherllin-Pirscher et al. (2017).

Another possible source of the absence of the patterns in RO monthly-means resides in the relative orientation of wave phase surfaces to be detected and the line-of-sight of the radio-occultation rays between transmitter and receiver satellites (Hierro et al., 2018; Alexander et al., 2008, Fig. 2 therein). This relative geometry varies considerably and quite weakens the sensitivity of RO to the wave structures, then on top damped by the monthly-mean smoothing effect. Moreover, the ideal-gas (equation of state) and hydrostatic approximations, intrinsic in RO pressure profiles retrieval, are no longer valid in these wave-perturbed domains, contributing another basic smoothing effect (Steiner and Kirchengast, 2000; Scherllin-Pirscher et al., 2017). In summary these properties imply that RO limitations in high space-time resolution turn out as an advantage in more robust accuracy achieved in longer-term larger-scale averages as used in this study.

Notwithstanding these properties, we emphasize at the same time that since the pioneering studies of Steiner and Kirchengast (2000) and Tsuda et al. (2000) many studies have proven the high value of RO data for gravity waves analyses (e.g., de la Torre and Alexander, 2005; de la Torre et al., 2006; Hierro et al., 2018). However, these mainly focused on vertical wave propagation and the well-resolvable aspects, and on extracting wave anomalies rather than averaging them out.

By testing the long-term stability of the difference between RO and ERA5 geostrophic winds from using long-term trend fits over 2007–2020, we revealed for certain latitudinal bands a trend magnitude of more than 0.5 m s$^{-1}$ per decade (Table 2). A change in the bias between the two datasets is especially visible after the year 2016, where major observing system changes occurred in the observational input data assimilated into ERA5 (Hersbach et al., 2020; Fig. 4 therein). This result indicates the potential advantage of RO-derived winds in terms of long-term stability for multi-decadal wind field monitoring, for example, to monitor the changes in large-scale circulation patterns such as the tropical-subtropical Hadley circulation (e.g., Weatherhead et al., 2018) or in the subtropical and polar jet streams, respectively.

This study advances on earlier initial studies to derive wind fields based on RO data (Scherllin-Pirscher et al., 2014, 2017; Verkhoglyadova et al., 2014). By comparing actual ERA5 wind and RO geostrophic wind, Scherllin-Prischer et al. (2014) commented that the difference is mainly caused by the wind approximation used, compared to the effect of RO retrieval errors. Our results corroborate these initial findings as we also found that the geostrophic approximation bias (against actual winds) is larger than the RO-ERA5 geostrophic winds difference. Beyond this, we advanced on several essential aspects in this study. The finer horizontal resolution used here (2.5°) compared to those of previous studies (5°), allowed us to go more equator-ward to reliably explore the region of the breakdown of the geostrophic approximation. While previous studies excluded





tropical regions between ±10° or ±15° (based on the argument of Coriolis force becoming small), we found that it is reliably possible to only exclude the ±5° equatorial band.

In addition, compared to the earlier studies, where few specific years were selected for the initial analyses, here we analysed long-term wind speed means, and the decadal-scale temporal stability, which gave more robust results. Additionally, the
345 difference between RO-derived geostrophic winds and actual winds from ERA5 was here clearly decomposed into one component from the geostrophic wind approximation bias against actual winds, and a separate one estimated from the difference between observational RO-derived geostrophic winds and ERA5-derived ones.

## 5 Conclusions and perspectives

The goal of this study was twofold. In a first step, we investigated the justification of using the geostrophic balance as an approximation for actual monthly-mean wind speed fields. We tested the validity of the geostrophic approximation both horizontally and vertically over the global domain, from the near-surface troposphere up to the middle stratosphere based on ERA5 reanalysis data. The successful performance of the method is of great importance for enabling, in principle, a reliable long-term dynamical wind field monitoring based on thermodynamic mass field data, such as available in form of RO-derived
isobaric geopotential height data. The geostrophic method is then a simple way to consistently derive wind fields from such geopotential height data. Hence, in a second step, we tested how well geostrophic winds obtained from RO data agree, with the ones estimated from ERA5 reanalysis data.

We found a near-global validity of the geostrophic approximation and clear indication of added value due to long-term stability of RO-derived winds, which highlights the potential of long-term RO-based monthly-mean wind field records for supporting
climate monitoring and analyses.

In particular, main findings include:

- the geostrophic approximation is valid over areas from ±5° poleward, in the troposphere and lower stratosphere (up to 10 hPa), with wind speed differences generally within ±2 m s$^{-1}$,

- larger ageostrophic contributions occur in the winter stratosphere (due to neglecting curvature effects) and over large
mountain ranges (due to the neglected gravity waves effects),

- the differences between RO and ERA5 geostrophic winds are generally (even) smaller than the residual biases due to the geostrophic approximation, with differences well within ±2 m s$^{-1}$,

- overall, the total difference between RO-derived wind and ERA5 actual wind is small in monthly-mean wind fields, outside a ±5° equatorial band and up to near ±85° at the polar caps,

- orographic gravity wave effects in monthly-mean geopotential height fields are found in ERA5 but not RO data, due to less susceptibility of RO to these smaller-scale perturbations,



- the ERA5 and RO long-term difference is exceeding the GCOS 0.5 m s$^{-1}$ per decade stability requirement in several latitude bands within 10° to 40° (up to ~1 m s$^{-1}$ per decade in 2007–2020 difference trends), pointing to inhomogeneity in ERA5 data due to observing system changes and potential added-value from the long-term stability of RO-derived wind field records.

Despite this decent progress towards assessing the utility of RO records for wind monitoring, some problems and questions remain. One of the problems is related to the wind information in the tropics close to the equator, in the ±5° equatorial band. Even though this region, where the geostrophic approximation breaks down is quite narrow, the question remains how to possibly "fill" it with valid (monthly-mean) wind information. Oberheide et al. (2002) used a linear interpolation of the geostrophic wind to estimate wind fields between ±10° latitude. Healy et al. (2020) showed that RO-derived zonal mean balance winds well quantify stratospheric zonal winds at the equator. However, this equatorial-balance-approximation approach did not provide information on geographically gridded wind fields and appears to lack information towards lower altitudes into the troposphere. Consequently, in follow-on research, one focus will be to help advance on this problem. Building on Healy et al. (2020) and Scaife et al. (2000), we will assess the utility of the balance-approximation for reasonably bridging this "equator gap" by RO-based wind fields. Another focus we pursue is a more detailed analysis of the separate zonal and meridional wind component fields.

Regarding improvements to the accuracy of RO-derived wind fields at altitudes higher than around 30 km (or 10 hPa) into the upper stratosphere, there are several approaches. One of the possibilities to decrease residual biases these high altitudes is to extend the geostrophic to the gradient-wind balance approximation to estimate wind in those regions. This might also help to close the "polar cap gap", i.e., latitudes beyond 82.5° that we did not cover by RO data in this study. Scherllin-Pirscher et al. (2014) showed that the geostrophic approximation was performing better in the region of the sub-tropical jet, while for the polar (night) jet the gradient-wind balance approximation performed better. Another avenue is the also-mentioned lower accuracy of RO data above about 30 km, mainly related to residual ionospheric biases. The potential of improving geopotential height data from RO at these altitudes does hence exist (e.g., Healy and Culverwell, 2015; Danzer et al., 2020, 2021; Liu et al., 2021; Syndergaard and Kirchengast, 2022), and the use of newest reprocessed RO data records is hence expected to also help improve wind monitoring in the upper stratosphere.

**Author contribution**

Conceptualization: GK, JD; Data curation: IN; Formal analysis: IN, JD; Funding acquisition: JD; Investigation: IN, JD; Methodology: IN, JD, GK; Supervision: JD, GK; Validation & Visualisation: IN, JD, GK; Writing – original draft preparation: IN, JD; Writing – review & editing: JD, GK.



**Acknowledgments**

We thank the UCAR/CDAAC RO team for providing RO excess phase and orbit data and the WEGC RO team for providing the OPSv5.6 retrieved profile data. We particularly thank F. Ladstädter (WEGC) for providing the monthly gridded climatology data and related discussions. Furthermore, we thank the ECMWF for providing access to the ERA5 reanalysis data. Finally, we thank the Austrian Science Fund (FWF) for funding the work; the wind analysis is part of the FWF stand-alone project Strato-Clim (grant number P-40182).

**Data Availability Statement**

The ERA5 data on pressure levels can be downloaded at (https://cds.climate.copernicus.eu/cdsapp#!/dataset/reanalysis-era5-pressure-levels?tab=form). The OPSv5.6 data are available at the website (https://www.doi.org/10.25364/WEGC/OPS5.6:2020.1).



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
