# Peer review of "Validation of the geostrophic approximation using ERA5 and the potential of long-term radio occultation data for supporting wind field monitoring"

_Atmospheric Measurement Techniques, 2023_

## Editor Comment (EC1)

Validation of the geostrophic approximation using ERA5 and the potential of long-term radio occultation data for supporting wind field monitoring

by Irena Nimac, Julia Danzer and Gottfried Kirchengast

The manuscript aims to validate the use of ERA5 reanalysis data and radio-occultation (RO) data fo long-term monitoring of wind fields. First the geostrophic relation is validated by comparing long-term averages of ERA5 wind speed data with geostrophic wind speed data determined from ERA5 geopotential data and, secondly, geostrophic wind speed derived from radio occultation geopotential data are compared with the ERA5 geostrophic winds. The manuscript concludes that RO determined wind data can provide added value to ERA5 wind data, for the benefit of wind field monitoring.

The manuscript is quite well written. I have one basic principal hesitation. As I understand it, and it is also stated on line 44 of the manuscript, RO observation data are assimilated into operational analysis for forecasting purposes as well as into reanalysis. In this case ERA5 data and RO data are dependent and should not be used to validate the potential of RO data to support wind field monitoring. It may even be so that RO data is the dominating observation data source over else data-void areas like oceans. The differences between Figures 7e (Northern Hemisphere winter) and 7f(Southern Hemisphere) support this criticism.

Other views

- Line 13: Also the use of ERA5 wind analysis data to compare and validate ERA5 geostrophic wind data based on ERA5 geopotential data is questionable. The ERA 5 wind and geopotential analysis increments are coupled via near-geostrophic linear relations. For this reason ERA5 wind and geopotential data are deterministically dependent, although the long term mean increment may be very small.

- Line 73: In my view, in case RO observations were used already, they cannot provide any further "added value".

- Line 80: It is stated that a latitude-longtude grid of 2.5 degrees resolution is used for ERA data. Please inform whether 2.5.degrees is also the resolution of the input spectral ERA5 data. Furthermore a latitude-longitude grid is not optimal for calculation of geostrophic winds in polar areas. Alternative Grids should be considered, at least in polar regions.

Lines 104-108: The sentence "RO data show …….. 2019)". is a bit un-clear.

---

## Author Comment (AC1)

**Author's Response to Referee #1**

We would like to thank the reviewer for helpful comments. Our answers to the comments are written in *italic*.

**Reviewer #1 (Comments to Author):**

1) It would be useful for the reader to better explain how the geopotential is obtained from radio occultation (RO) data, as it is not a quantity obtained directly from RO observations, but derived from the integration of atmospheric density.

*We agree that it may be useful if we refine this part a bit. We will hence add a short text like the following (first draft) in the "Data and study method" part in the revised version.*

*"Based on the atmospheric bending of the GNSS signals during the occultation sounding, it is possible to retrieve atmospheric refractivity profiles. From these, air density and pressure profiles as a function of altitude, or geopotential height, can be accurately derived based on the refractivity equation, the equation of state, and the downward integration of the hydrostatic equation. In this way, geopotential height profiles as a function of pressure levels can be obtained with unique accuracy and form the basis for the wind field derivation (for a more detailed description see Scherllin-Pirscher et al., 2017 - https://doi.org/10.1002/2016JD025902)."*

2) Please indicate the major changes that occurred in 2016 in the ERA5 observing system that may explain the changes observed in the difference between ERA5g and ROg.

*These changes are mainly related to the changes in the certain input data into the data assimilation system that produced the reanalysis. To better indicate this, we will add a short text like the following (first draft) to the "Discussion" part in the revised version.*

*"A change in the bias between the two datasets is especially visible after the year 2016, where certain observing system changes occurred in the observational input data assimilated into ERA5 (Hersbach et al., 2020). Specifically, a salient increase in the number of assimilated observations is seen around this year for surface pressure and specific humidity (Hersbach et al., 2020; Fig. 3 therein). In addition, the inclusion of WIGOS AMDAR (WMO Integrated Global Observing System, Aircraft Meteorological Data Relay) data in 2015 and the exclusion of some wind profilers and ACARS (Aircraft Communications Addressing and Reporting System) in 2016 are likely further sources of the inhomogeneities (Hersbach et al., 2020; Fig. 4 therein)."*

3) In the winter middle. stratosphere (10hPa), the geostrophic approximation overestimates the wind speed and the overestimation seems to increase with wind speed. It is greater in the SH where the stratospheric jet is faster than in the NH. If the cyclostrophic term is taken into account (term in $V^2/R$ due to the rapid circulation around the polar vortex), does the agreement with the ERA5 winds become better?

*Thank you – but yes, we comment in the discussion part of the manuscript that these larger differences at higher levels (i.e. middle stratosphere) of the winter hemisphere are also a result of neglecting this important cyclostrophic term (i.e., using the geostrophic approximation instead of the gradient wind approximation; cf. e.g., Scherllin-Pirscher et al., 2014). Here, we did not want to go into the details of really including gradient wind estimates also, since we wanted to focus on the validity of the geostrophic approximation, but indeed we do plan to combine various approximations in the future studies (also the so-called equatorial balance approximation across the equator; cf. Danzer et al., AMTD, in review, 2023).*

4) Please check the alphabetical order of the publications (i.e. Hierro before Healy)

*Thank you for noticing this. We will correct this in the revised version of the manuscript.*

---

## Author Comment (AC2)

**Author's Response to Referee #2**

We would like to thank the reviewer for the thorough and helpful comments. Our answers to the comments are written in *italic*.

**Reviewer #2 (Comments to Author):**

Initial assessment: The paper makes use of ERA5 monthly mean geopotential fields for July and January on pressure levels from 1000 hPa to 10 hPa and 2.5°x2.5° horizontal grid during the time period 2007 to 2020. For the same time period and resolution, monthly mean geopotential is computed from RO data using aggregated atmospheric profiles. From each geopotential data set, local geostrophic wind is computed using the geostrophic wind equation at pressure levels. The ERA5 and RO geostrophic winds are then used to "1) test the validity of the geostrophic approximation for representing monthly-mean winds, and to 2) evaluate the utility of RO derived monthly-mean winds, for their potential added value as a separate wind field monitoring data record providing improved long-term stability".

1) The paper reads as a technical report and provides little scientific understanding.

*Yes, the paper has a focus on methodology of atmospheric processing of RO data (which is why we submitted it to AMT and not, e.g., ACP), but we also discuss at various suitable places the scientific implications, in the sense of pointing to the scientific utility of the data for climate monitoring. We agree that we do not focus on analyzing and understanding at the same time atmospheric processes as such, since we consider it would overstretch the scope of this AMT paper. As we try to say clearly from the introduction onward (and we will aim to make this even more clear in the revised version), the main goal of the study is to evaluate the potential to derive geostrophic wind fields from radio occultation satellite measurements, and the geographical and altitudinal scope of their utility. This is a quite new topic within the RO community, as it did not get a lot of attention so far (we of course aim to carefully cite all existing suitable references). Hence, our approach is to first evaluate the quality of the derived RO wind fields, which this paper sent to AMT focuses on. Once the quality is established, scientific applications in the field of atmospheric and climate physics and dynamics are possible, and we would then focus on these. To cite a recent example of this publication approach related to our team, please see Li et al. AMT (2021) (https://doi.org/10.5194/amt-14-2327-2021 - focus methodology) and Li et al. ACP (2023) (https://doi.org/10.5194/acp-23-1259-2023 - focus physics/dynamics).*

2) The method of the computation of geopotential from RO data is not well explained, e.g. it is unclear if it is only the hydrostatic height from temperature profile or also moisture was accounted for, whether averaging was carried out on input RO profiles or derived geopotential.

*Ok, thank you, we agree that it will be useful to provide more refinement of the description to this end. We hence plan to add a short text like the following (first draft) in the "Data and study method" part in the revised version.*

*"Based on the atmospheric bending of the GNSS signals during the occultation sounding, it is possible to retrieve atmospheric refractivity profiles. From these, air density and pressure profiles as a function of altitude, or geopotential height, can be accurately derived based on the refractivity equation, the equation of state, and the downward integration of the hydrostatic equation. In this way, geopotential height profiles as a function of pressure levels can be obtained with unique accuracy and form the basis for the wind field derivation (for a more detailed description see Scherllin-Pirscher et al., 2017 - https://doi.org/10.1002/2016JD025902)."*

3) More importantly, the computation of geostrophic winds is in the opinion of this reviewer unsuitable. The authors compute the geostrophic wind on the local f plane, latitude by latitude. Instead, one should derive geostrophic winds on the sphere that provide a smooth representation of the flow. An appropriate way of doing this is by using the stream function, ideally on model levels to avoid any errors due to interpolation, especially over the orography and in the lower troposphere. This global stream function provides the geostrophic wind on the sphere including the tropics.

On the other hand, this does not allow the comparison with the RO data in terms of geostrophic winds. If this is what the authors really intend to do as argued by their goal 2), the analysis should be described as "... geostrophic winds on a local f-plane..." and limited to the midlatitude free troposphere region.

*Thank you, yes, we should make (even) more clear, that in line with predecessor work in the field (cited as references), that we apply the geostrophic approximation in a local-regions sense, i.e., evaluating the respective derivative equations locally on geopotential height fields on isobaric levels, where the horizontal sampling is of order 2.5° x 2.5° (and resolution of order 5° x 5°). Actually, in one more recent of our predecessor works, the paper by Scherllin-Pirscher et al. (2017 - https://doi.org/10.1002/2016JD025902), we alternatively derived the wind field as the gradient vector field of the Montgomery potential at potential temperature surfaces, as well as, for comparison, from the geopotential at isobaric surfaces, as we do in this study. The numerical results were found essentially identical (at ~2.5° sampling with ~5° resolution); and the geopotential-based derivation method has some advantages, since it makes like-to-like comparison of results from RO data to reanalysis data quite straightforward. In summary, we do understand that we need to improve the introduction of our method of computation, so that it more clears also to readers that come more from the dynamical meteorology side, and that we did base on the "local f plane" conception. And also recheck we are really clear that we apply the geostrophic approximation on ERA5 data in the same way we apply it to RO data, to fully understand the ageostrophic contribution.*

4) If the authors choose to resubmit the manuscript, more consideration should be given to the interpretation. The differences between the full and geostrophic winds should be called ageostrophic wind (not a "bias"), etc.

*Yes, ok, we would strive to improve on the interpretation side, as summarized above; however we will prefer to keep it as a methodology-focsed paper here for AMT, since otherwise the scope would become too broad and the paper too long. Specifically, we also agree with the suggestion to carefully check, and as needed rectify, the use of the word "bias", both in cases where it's actually just a*

*"systematic difference" (and no one dataset can be set biased against the other) and also in physical interpretation contexts, like if we actually should correctly say „ageostrophic wind".*

---

## Author Comment (AC3)

**Author's Response to Referee #3**

We would like to thank the reviewer for the valuable and helpful comments. Our answers to the comments (quoted in gray) are written in *italic* below each of the comments.

**Reviewer #3 (Comments to Author):**

Initial assessment: The manuscript aims to validate the use of ERA5 reanalysis data and radio-occultation (RO) data for long-term monitoring of wind fields. First the geostrophic relation is validated by comparing long-term averages of ERA5 wind speed data with geostrophic wind speed data determined from ERA5 geopotential data and, secondly, geostrophic wind speed derived from radio occultation geopotential data are compared with the ERA5 geostrophic winds. The manuscript concludes that RO determined wind data can provide added value to ERA5 wind data, for the benefit of wind field monitoring.

1) The manuscript is quite well written. I have one basic principal hesitation. As I understand it, and it is also stated on line 44 of the manuscript, RO observation data are assimilated into operational analysis for forecasting purposes as well as into reanalysis. In this case ERA5 data and RO data are dependent and should not be used to validate the potential of RO data to support wind field monitoring. It may even be so that RO data is the dominating observation data source over else data-void areas like oceans. The differences between Figures 7e (Northern Hemisphere winter) and 7f (Southern Hemisphere) support this criticism.

*Thank you for basically finding our manuscript quite well written; we aimed to do everything with due care, science-wise and in the writing.*

*Regarding your "one principal hesitation", we recognize that we apparently have not been clear enough in the introduction of our study design and purpose. We hence will improve on this in the revised manuscript, at several places both in the introduction and the method description, and in fact also plan to improve our what-we-do-term in the paper title from "Validation of…" to "Evaluation of…". On the latter, we realize that we may misguide some readers with using the term "Validation" for actually an evaluation design, where we 'just' test approximations; one physics-approximation (use of local geostrophic equations vs. of full dynamics equations to obtain U,V) and one sampling-approximation (use of 'a-bit-sparse' observational RO sampling vs. of 'full-coverage' space-time-gridded (re)analysis sampling). Given this, the essence of this initial response to your query is that the results of both steps of our "two-steps evaluation method" (quoted from the caption of Figure 1 that introduces our two goals) do very weakly depend on which (re)analysis we use as a reference, as long as it is a state-of-the-art (re)analysis. That is, similar to the cases where we estimated "under-sampling biases" (also termed "sampling errors") and other approximation biases in previous climate-related evaluation studies for various RO variables (e.g., Scherllin-Pirscher et al., 2011, 2014, 2017; https://doi.org/10.5194/amt-4-2019-2011, https://doi.org/10.1002/2014GL061524, https://doi.org/10.1002/2016JD025902), a sound validity of the estimated "approximation-vs.-'true'-reference" bias fields requires a state-of-the-art (re)analysis that represents quasi-realistic atmospheric variability at sub-daily to multi-month time scales at 100-km-scale to large-scale spatial resolution. We most often used ECMWF (re)analyses and/or short-range forecasts (e.g., Steiner et*

*al. 2013, [https://doi.org/10.5194/acp-13-1469-2013](https://doi.org/10.5194/acp-13-1469-2013); Ladstädter et al. 2023, [https://doi.org/10.1038/s41598-023-28222-x](https://doi.org/10.1038/s41598-023-28222-x)), but also others were tested. We agree that (re)analysis or short-range forecast fields from assimilation-forecasting systems where RO data would not be assimilated, would be somewhat less quasi-realistic in representing the atmospheric variability in the mass and wind fields, indeed reflecting the positive impact of the assimilated thermodynamic RO data information content. However, since all major state-of-the-art (re)analyses do assimilate RO data since 2006 (start of the "U.S. COSMIC" and "European Metop" RO multi-satellite era), we consider it adequate in this study to quantitatively evaluate the validity of the geostrophic approximation (first goal), and of the RO-sampling sufficiency for providing adequate monthly isobaric-level geopotential fields (second goal), based on using the representative mass and wind field data of the state-of-the-art reanalysis ERA5 (from other studies also involving short-range forecasts, or MERRA2, JRA-55 reanalyses like in von Schuckmann et al. 2023 [https://doi.org/10.5194/essd-15-1675-2023](https://doi.org/10.5194/essd-15-1675-2023), Section 3 therein, we do not expect major differences for the present purpose; we agree, though, we could also test this for a few months for this study, which we so far considered an effort beyond the scope of this initial study). In summary, we hence consider the indirect "dependence" of ERA5 on RO data via assimilation of (some of) the latter data fully compliant with the goals of this study that is actually not including validation in the more strict sense of the term (sorry for the terminological mislead that we will rectify); we plan to improve the revised manuscript to make this clearer.*

Other views:

2) Line 13: Also the use of ERA5 wind analysis data to compare and validate ERA5 geostrophic wind data based on ERA5 geopotential data is questionable. The ERA 5 wind and geopotential analysis increments are coupled via near-geostrophic linear relations. For this reason ERA5 wind and geopotential data are deterministically dependent, although the long term mean increment may be very small.

*Thank you, please see the answer above. We agree that the 3D-gridded U,V wind field results of ERA5 outputs, obtained from the ECMWF model dynamics&physics underlying the data assimilation in ECMWF's integrated forecasting system (IFS), will (hopefully!;) be physically consistent, hence "deterministically dependent", with the isobaric-level geopotential fields at the appropriate space-time scales. But as you hint, we focus in this study on monthly-means-based long-term wind field monitoring, at highest horizontal resolution no finer than several 100 km (2.5° x 2.5° sampling, order 5° lat x 5°/cos$\varphi$ lon resolution), i.e., we look at quite strongly space-time-filtered wind fields, where most of ageostrophic components are attenuated quite strongly. We consider ERA5, and the underlying ECMWF IFS, in the context a state-of-the-art atmospheric analysis asset that is capable to provide us with quasi-realistic dynamics&physics, properly bridging from 6-hourly analyses per day to monthly-mean fields at the synoptic- to large-scale horizontal resolution of interest.*

3) Line 73: In my view, in case RO observations were used already, they cannot provide any further "added value".

*Thank you, your view indicates that we may also not have made it sufficiently clear so far that the key phrase for the "added value" is, as explicitly included in the title, "…the potential of long-term radio occultation data…". That is, what RO may deliver as "added-value" based on its rather unique*

*combination of high accuracy and long-term stability (=multi-year to multi-decadal stability), is the capacity to accurately keep long-term consistency also over certain months and times where reanalyses (like ERA5) experience inhomogeneities due to changes in observing systems, i.e., in the combination of assimilated data sets in observation type, amount, quality, and space-time coverage. We in fact found in this study indication of one such "inhomogeneity year" (2016) in ERA5. A more long-term stable monthly-mean RO data record might hence cover, say, the 15 years 2006 to 2020 wind-field record more stably than the corresponding record from ERA5, which would be an added-value for climate change-related studies like long-term gradual shifts of the Hadley cell, jet stream patterns, etc. We will carefully recheck the text, to make sure we improve the description of what "added-value for long-term monitoring" means, where found needed. By the way, a good example for such added value was recently published in a climate change-related study based on RO-derived long-term temperature fields (Ladstädter et al. 2023, [https://doi.org/10.1038/s41598-023-28222-x](https://doi.org/10.1038/s41598-023-28222-x)).*

4) Line 80: It is stated that a latitude-longtude grid of 2.5 degrees resolution is used for ERA data. Please inform whether 2.5.degrees is also the resolution of the input spectral ERA5 data.
Furthermore a latitude-longitude grid is not optimal for calculation of geostrophic winds in polar areas. Alternative Grids should be considered, at least in polar regions.

*Yes, the ERA5 data are from T42 spectral, consistent with the 2.5° grid (and various settings in the technical fine details how to do this, based on the original model level fields, have been tested very carefully as part of various previous studies). Regarding RO, we do in fact use equal-area cellsize selection around each grid-cell center location; hence the 2.5° x 2.5° sampling grid does not imply that we get, due to meridian convergence, smaller-and-smaller cell areas towards the poles. But thanks for mentioning this; we will recheck our description of these method details and improve as needed.*

5) Lines 104-108: The sentence "RO data show …. 2019)". is a bit un-clear.

*Ok, thanks, just looked at it – agreed, yes, we need to polish this one a bit to make it more clear…we will do so.*